# Towards Multiplexed and Multimodal Biosensor Platforms in Real-Time Monitoring of Metabolic Disorders

**DOI:** 10.3390/s22145200

**Published:** 2022-07-12

**Authors:** Sung Sik Chu, Hung Anh Nguyen, Jimmy Zhang, Shawana Tabassum, Hung Cao

**Affiliations:** 1Department of Biomedical Engineering, Henry Samueli School of Engineering, University of California Irvine, Irvine, CA 92697, USA; sungsc@uci.edu (S.S.C.); jimmyz10@uci.edu (J.Z.); 2Department of Electrical Engineering and Computer Science, Henry Samueli School of Engineering, University of California Irvine, Irvine, CA 92697, USA; hunganhn@uci.edu; 3Department of Electrical Engineering, College of Engineering, The University of Texas at Tyler, Tyler, TX 75799, USA

**Keywords:** metabolic syndrome, diabetes, cardiovascular disease, obesity, inflammation, adipokines, multiplexed sensor, electrochemical sensor, optical sensor, point-of-care

## Abstract

Metabolic syndrome (MS) is a cluster of conditions that increases the probability of heart disease, stroke, and diabetes, and is very common worldwide. While the exact cause of MS has yet to be understood, there is evidence indicating the relationship between MS and the dysregulation of the immune system. The resultant biomarkers that are expressed in the process are gaining relevance in the early detection of related MS. However, sensing only a single analyte has its limitations because one analyte can be involved with various conditions. Thus, for MS, which generally results from the co-existence of multiple complications, a multi-analyte sensing platform is necessary for precise diagnosis. In this review, we summarize various types of biomarkers related to MS and the non-invasively accessible biofluids that are available for sensing. Then two types of widely used sensing platform, the electrochemical and optical, are discussed in terms of multimodal biosensing, figure-of-merit (FOM), sensitivity, and specificity for early diagnosis of MS. This provides a thorough insight into the current status of the available platforms and how the electrochemical and optical modalities can complement each other for a more reliable sensing platform for MS.

## 1. Introduction

Metabolic syndrome (MS) refers to a cluster of metabolic disorders resulting from several impaired biochemical pathways [1,2,3]. These disorders include obesity, high blood pressure, high blood sugar, and abnormal cholesterol. According to the American Heart Association (AHA), approximately 35% of adults and 50% of people aged 60 years and older suffer from MS in the United States alone. On a global scale, nearly 13% of adults aged 18 years and older had obesity in 2016, according to World Health Organization (WHO). People with MS are at an increased risk of mortality from cardiovascular complications, stroke, and chronic kidney diseases. With the global public health and economic crises created by the severe acute respiratory syndrome coronavirus 2 (SARS-CoV-2), obesity and dysfunctional metabolism also affect the course of Coronavirus disease 2019 (COVID-19) [4]. Importantly, clinical studies show that obesity and impaired metabolic health are independent determinants of severe COVID-19.

The exact pathogenesis of MS is not well understood. However, research has demonstrated that MS is a chronic inflammatory state resulting from dysregulation of the immune system activity. Adipose tissues and the liver are the key inducers of chronic inflammation in MS. Overproduction of inflammatory adipokines by adipose tissue plays a key role. Adipokines are a group of up to 600 bioactive molecules, which primarily include leptin, adiponectin, monocyte chemotactic protein-1 (MCP-1), pro-inflammatory cytokines (e.g., Interleukin-6 (IL-6), Interleukin-1 (IL-1), and tumor necrosis factor-alpha (TNF-α)), among others [1]. Overexpression of adipocyte-derived MCP-1 recruits monocytes that secrete TNF-α, IL-6, and MCP-1. Additionally, IL-6 and TNF-α upregulate MCP-1 signaling, resulting in repeated cycles of the inflammatory cascade and self-perpetuating adipose tissue inflammation [1]. The liver expresses IL-6 receptors (IL-6R) and adiponectin receptors, which interact with IL-6 and adiponectin molecules derived from adipocytes [1]. TNF-α is also produced by the liver. Hence, the inflammatory burst generated by the adipose tissues triggers a subsequent cycle of inflammation in the liver, perpetuating chronic inflammation. The primary risk factors associated with MS include hyperglycemia, hypertension, insulin resistance, elevated total cholesterol, high BMI (body mass index), increased levels of LDL (low-density lipoprotein), decreased levels of HDL (high-density lipoprotein), elevated triglycerides, and oxidative damage. The pro-inflammatory cytokines (i.e., IL-6, IL-1, and TNF-α), the chemokine MCP-1, leptin, and uric acid are positively correlated with MS risk factors. In contrast, anti-inflammatory cytokines (e.g., IL-10) and adiponectin are negatively associated with the risk factors, as depicted in Figure 1 [3]. Several studies demonstrate that individuals with these aforementioned risk factors are at an increased risk of severe cardiovascular complications, type II diabetes, renal and liver failure, stroke, vascular dysfunction, and in extreme cases, cancer. Therefore, early detection and quantification of these circulating biomarkers will facilitate the early intervention of metabolic disorders and prevent the resultant adverse clinical outcomes.

MS is a complex and multi-factorial condition comprising a cluster of simultaneous conditions that increase the risk of type 2 diabetes and heart disease. The exact definition of MS varies among organizations, including the World Health Organization (WHO), the National Cholesterol Education Program (NCEP), and the International Diabetes Federation (IDF). The most widely accepted definition among them is the criteria set by NCEP and IDF, which use waist circumference, the level of cholesterol, blood pressure, and fasting plasma glucose [5]. However, MS diagnosis with these parameters does not necessarily specify the disorder that the patient has, and they need to go through further clinical testing for precise diagnosis. The diagnostic procedures that follow after the general MS diagnosis may differ depending on which disorder the patient is suspected to have, and they most likely require time-consuming and laborious steps, such as blood testing, genomic testing, imaging, etc. This necessitates the need for a platform that can rapidly diagnose MS. To date, sensors have been developed for this purpose to identify independent conditions only [6]. Sensors that detect a single analyte may not provide conclusive results for diagnosing diseases associated with multiple complications, such as MS. Hence, there is a need to develop a multi-analyte sensing platform to monitor the inflammatory pathways for studying the interplay of molecular biomarkers, analyzing the progression of risk factors associated with MS, and conducting a personalized risk assessment. Diagnostic and screening tools that provide results in minutes and monitor multiple circulating biomarkers in real time will enable rapid implementation of control measures and facilitation of timely treatment. In this review article, we first summarize the biomarkers related to different types of MS and the potential of non-invasively accessible biofluids for detecting the circulating biomarkers associated with MS. Next, we discuss two widely used transduction mechanisms, the electrochemical and optical methods, in developing multiplexed point-of-care (POC) sensing platforms for early diagnosis of MS and monitoring its progression (Figure 2). The knowledge stemming from this review will help researchers outline a path forward to fight the evolution of metabolic disorders and the resulting health complications.

## 2. Biomarkers for Metabolic Syndrome

POC devices that continuously monitor the physiological signals of our bodies can provide unique insights into our health [7]. The conventional tests conducted in a clinical setting occur only a few times a year, whereas POC devices offer continuous access to real-time physiological data. By performing near-patient testing with a small samples using portable devices, POC provides fast results with reliable diagnosis [8]. Biochemical markers are useful indicators for the progression of various health conditions, including stress, infection, and cardio-pulmonary abnormalities [9,10,11]. Serial measurements of specific proteins have proven advantageous in determining adverse clinical outcomes and death in critically ill MS patients [12,13,14]. However, the blood test is currently the only established and standardized method of measuring these biomarker levels. Blood tests are invasive, have an average turnaround time of several hours to days, and are difficult to conduct frequently to monitor disease progression. Thus, they impede the identification of dynamic biomarkers and significantly increase the risk of infections through cross-contaminations, which are life-threatening for infants and critically ill patients (Figure 3). Additionally, the current procedure is not feasible in rural areas with limited standard protocols and trained personnel [15]. The diagnostic potential of body fluids, such as sweat, tears, and breath, is significantly underutilized. Advances in technology have enabled access to previously inaccessible biochemicals with minimum discomfort and supervision. In this section, we will discuss the possible biomarkers that can be used for diagnosing MS (Table 1). The subsequent sections discuss the utility of five non-invasively accessible biofluids (sweat, tears, breath, saliva, and urine) in detecting and monitoring protein biomarkers along with emerging low-cost and multiplexed POC sensors that are expected to substantially improve diagnostic and prognostic outcomes in MS patients. 

### 2.1. Metabolic Biomarkers for Predicting Cardiovascular Disease

Cardiovascular diseases (CVDs), a group of diseases or conditions that affect the heart or blood vessels, including arrhythmia, aorta disease, heart attack, heart failure, cardiomyopathy, etc., are among the most prevalent MS in the world [16,17]. Cardiovascular disease is a severe condition that is considered the leading cause of death worldwide. Most CVDs are related to atherosclerosis, where a plaque builds up in the arteries and blocks blood flow. Although several behavioral patterns, such as obesity, smoking, frequent alcohol consumption, and lack of physical activity are thought to increase the probability of CVDs, biomarkers such as C-reactive protein (CRP), troponin I (cTnI), procalcitonin (PCT), to name just a few, also play a role in the development of CVDs [18].

Factors such as cholesterol, low-density lipoprotein (LDL) cholesterol, high-density lipoprotein (HDL) cholesterol, and triglycerides are considered conventional biomarkers for predicting CVDs. These have been used widely through the years for risk prediction models and diagnosis. However, recent studies showed that these traditional biomarkers are insufficient [19]. They are not always expressed, and, in some cases, none or only one of these biomarkers are observed in patients with CVDs [20]. Thus, novel biomarkers are being investigated for enhanced CVD risk management. Since various conditions are related to CVDs, the biomarkers for CVDs can also be classified according to the associated conditions, such as myocardial necrosis, inflammation, plaque instability, platelet activation, myocardial stress, etc.

One of the widely studied biomarkers is cardiac troponin. Troponin is a protein found in the heart muscles that inhibits contraction through its interaction with actin and myosin. Normally, it is not present in the blood. However, when the heart is damaged, it is released into the bloodstream, making it a possible biomarker for myocardial injury. Current conventional assays for detecting cardiac troponin have limitations, including low sensitivity [19]. Another biomarker for CVDs is CRP. CRP is a protein that is upregulated when there is inflammation in the body. It is being used to determine the risk of coronary artery disease. The relationship between CRP and cardiovascular events has been studied extensively, and its direct association has been proved. Despite its relevance, it’s not a casual factor expressed in CVDs [19]. In summary, a vast number of biomarkers for CVDs have been studied. However, each biomarker holds its limitation, increasing the need for a multiplexed CVD sensing platform.

### 2.2. Metabolic Biomarkers for Prediabetes

Diabetes refers to conditions where the body is either incapable of producing enough insulin or of using the insulin that is produced. These two pathologies are classified as type 1 and type 2 diabetes, respectively. Type 1 diabetes, where insulin production in the body is stopped, is known to be caused by an autoimmune reaction of the body, and its prevention method is yet to be found. In contrast, type 2 diabetes with insulin resistance can be prevented by practicing a better lifestyle, such as losing weight and eating healthy [21,22]. Prior to type 2 diabetes, patients develop a condition called prediabetes. With this condition, the glucose level is higher than normal but is not considered to be diabetes, being between 100–125 mg/dL (5.6–6.9 mmol/L) [23].

As diabetes is one of the leading causes of serious health complications, including heart disease, vision loss, kidney failure, ulcers, etc., the field of diabetic biomarker detection has been investigated by many researchers. Glucose monitoring has been emphasized as an area of interest due to its direct relationship with diabetes. While several research groups have reported the association between blood glucose and other bodily fluids (such as sweat, tears, and saliva) glucose concentrations, the field of glucose monitoring has started to transition from utilizing invasive blood samples to noninvasive biofluids. 

However, glucose is not the only biomarker for diabetes. There are a variety of other biomarkers that are reported to be related, including the cluster of differentiation 14 (CD14), CD 99 serum amyloid (SA), HbA1c, glycated albumin (GA), oral glucose tolerance test (OGTT), adiponectin, FetA, etc. [24]. As the inflammatory response caused by obesity and unhealthy lifestyle is the main cause for diabetes, biomarkers that are related to inflammation are also being investigated. For example, HbA1c is a molecule that is produced when glucose attaches to hemoglobin. An HbA1c level between 39–46 mmol/mol is thought to indicate prediabetes. However, HbA1c alone has its limitations regarding sensitivity to diabetes diagnostics [25]. Hence, using it in conjunction with fasting plasma glucose (FPG) and oral glucose tolerance test (OGTT) improves the reliability of the results. Another biomarker that can be used is the monocyte differentiation antigen, CD 14, which has been reported for its effect on insulin resistance [24]. Likewise, several biomarkers are found to be related to either prediabetes or diabetes, increasing the need for a multimodal platform for diabetic diagnosis with higher accuracy.

### 2.3. Metabolic Biomarkers for Cancer

Cancer is a complex disease that involves cancer-associated genes, including approximately 250 oncogenes and 700 tumor suppressors [26]. These genes can regulate and interact with metabolic genes, which cause dysfunctional metabolism in different cancer genotypes. So far, 10,000–50,000 different single nucleotide variants of most oncogenes and tumor suppressors in tumor cells have been identified as playing a key role in three major cellular metabolism pathways [27]. Based on the type of cancer, the pathways of cellular metabolism include (i) aerobic glycolysis, (ii) glutaminyls, and (iii) one-carbon metabolism, needed for rapid cell growth and division. Most of the pyruvate formed from glycolysis enters the tricarboxylic acid (TCA) cycle in healthy cells and is oxidized via oxidative phosphorylation. However, in cancerous cells, pyruvate is mainly converted to lactic acid to respond to the requirements of the metabolic mechanisms for their continuous cell growth. The mechanisms regulate glycolytic enzymes expressed in cancer cells through multiple pathways; for example, normally allosterically inhibiting rate-controlling steps [28]. The pathway is associated with phos-phofructokinase (PFK), sustaining a high rate of glycolysis. Usually, this pathway is inhibited by ATP. However, when glucose is abundant, fructose 2,6-bisphosphate, a product of 6-phosphofructo-2-kinase/fructose 2,6-bisphosphatases, can override ATP-mediated PFK inhibition [29]. In cancer cells, elevated levels of fructose 2,6 bisphosphate are produced under hexokinase activity, which allosterically activates PFK [30]. As a result of the different mechanisms of activation, PFK activity is much higher in cancer cells than in normal tissue. 

Malfunctions in the metabolism of glucose, protein and carbohydrates, fatty acid oxidation, mitochondrial function, neurotransmitter metabolism, and markers of oxidative stress and microbiome contribute to cancer development. The concept of “oncometabolite” is associated with metabolic disorders. For instance, the accumulation of a high concentration of oncometabolites, including 2-hydroxyglutarate, fumarate, sarcosine, glycine, glucose, glutamine, serine asparagine, choline, lactate, and polyamines initiates tumor growth and metastasis [31]. The shunting of pyruvate to secreted lactase in cancer is associated with elevated levels of lactate dehydrogenase (LDH) and monocarboxylate transporter (MCT) that cotransport lactate and a proton out of the cell [32]. LDH and MCT as metabolites have been observed in several cancers in the tumor microenvironment. These enzymes are involved in metabolic pathways that fulfill elevated metabolic demands for bioenergetics and cellular biosynthesis. The shift toward lactate production and away from oxidative phosphorylation also reflects the decreased activity of the LDH complex, which can result from the induction of the inhibitory pyruvate dehydrogenase kinases (PDKs). In addition, changes in the expression level of metabolic enzyme isoforms and activities of metabolic enzymes, especially for increased secretion of lactate, cause metabolic disorders in cancer. 

These endogenous metabolites indirectly modify histone methylation patterns, which are associated with activation or repression of the transcription process for both oncogenes and tumor suppressors [33]. In addition, recent electron images and bioassay measurements displayed abnormal mitochondria structure, low numbers of mitochondria, and insufficient oxidative phosphorylation (Oxphos) in the tumor tissue of over 80% of cancer patients [34,35]. These losses are compensated by upregulation of glycolysis and lactate fermentation in order to sustain tumor cells. This would support Warburg’s hypothesis that respiration is insufficient in cancerous cells. 

Although recent methods have advanced cancer treatments, early cancer detection is still the best way to ensure effective treatment outcomes. These cancer metabolite biomarkers have been used for cancer detection based on simple metabolic changes, such as their increased levels in blood, saliva, breath, or urine. Some existing detection platforms for metabolic enzymes have shown significant potential in cancer detection [36]. Indeed, recent publications showed that genetic screening cannot detect more than 95% of cancer due to somatic origin [26]. Therefore, metabolite screening could be an alternative approach that provides fast, cost-effective, and early-stage detection of cancer. Advances in noninvasive image processing techniques, including positron emission tomography (PET), magnetic resonance spectroscopy, and metabolomic biosensor tests, can enable the detection of oncometabolites in different cancer phenotypes.

**Table 1 sensors-22-05200-t001:** Metabolic syndrome biomarkers and its relevant concentrations.

Metabolic Syndrome	Biomarker	Clinical Approval	Concentration	Refs.
Cardiovascular Diseases (CVDs)	C-reactive Protein (CRP)	Approved	>3 mg L^−1^	[37]
	Highly sensitive CRP		<1 mg L^−1^	[38]
	Cardiac troponin 1 (cTn1)	Approved	>0.5 μg L^−1^	[39]
	Procalcitonin		>67.89 μg L^−1^	
	Cholesterol	Approved	>240 mg dL^−1^	[40]
	LDL cholesterol	Approved	>130 mg dL^−1^	[41]
	HDL cholesterol	Approved	<40 mg dL^−1^	[42]
	Triglyceride		>150 mg dL^−1^	[43]
Diabetes	Glucose	Approved	>125 mg dL^−1^	[44]
	CD14			[45]
	CD99			[46]
	HbA1c	Approved	>6.5%	[24]
	GA		>16.9%	[47]
	Adiponectin		<6 mg mL^−1^	[48]
	Fructosamine	Approved	<2.5 mmol L^−1^	[24]
Cancer	Fumarate	Approved	>1.35 mcg mg^−1^ creatinine	[49]
	2-hydroxyglutarate	Approved	>700 ng mL^−1^	[50]
	Sarcosine	Approved	>5000 nM	[51]
	Polyamines	Approved	35 kU L^−1^	[52]
	Lactate	Approved	>1.8 mmol L^−1^	[53]
	Lactate dehydrogenase		>280 U L^−1^	[54]
Autoimmune disease	Hydrogen peroxide (H_2_O_2_), hydroxyl radical (OH), superoxide anion radical (O_2_^−^), and nitric oxide (NO)	(investigating)	308 ppb (cutoff of 77 nL mL^−1^)	[55]
	Serum fatty acids (monounsaturated fatty acids such as lauric acid (C12:0), myristic acid (C14:0), stearic acid (C18:0), lignoceric acid (C24:0), palmitic acid (C16:0) and heptadecanoic acid (C17:0); saturated fatty acids, Cis-10-pentadecanoic acid (C15:1), Cis-11-eicosenoic acid (C20:1n9), and erucic acid (C22:1n9) as well as the gamma-linolenic acid (C18:3n6) polyunsaturated fatty acid))	(investigating)	86.7% specificity (ROC analysis)	[56]
	Serum fatty acid (3-hydroxypropionic and methylcitric acids, propionylglycine, tiglylglycine, 3-hydroxy-n-valeric, and 3-keto-n-valeric acids)	investigating	0.856 (ROC analysis)	[56]

Abbreviations: LDL, low-density lipoprotein; HDL, high-density lipoprotein; CD, cluster of differentiation, HbA1c, glycated hemoglobin; GA, glycated albumin.

### 2.4. Metabolic Biomarkers for Autoimmune Disease

The immunometabolism of immune cells is of emerging interest in therapeutic implications. Recently, the association between metabolic disorders and autoimmunity has been of great interest for early diagnostics [57]. Discrete metabolic pathways contribute to the regulation of growth, differentiation, survival, and activation of immune systems by providing energy and specific activation ligands [58]. Recent reports showed the importance of significant metabolic alterations for immune homeostasis in allergic diseases and rheumatoid arthritis [59]. Metabolic regulation of mucosal barriers, antigen-presenting cells, CD4 T cells, nutritional status, the intestinal microbiome, and inflammatory pathologies contribute to the development of autoimmune diseases. 

As the first line in the immune system, the metabolic regulation of mucosal barriers is a key contributor to allergic responses through reactive oxygen species (ROS). ROS systems include hydrogen peroxide (H_2_O_2_), hydroxyl radical (OH), superoxide anion radical (O_2_^−^), and nitric oxide (NO), which play an important role in the elimination of bacterial and fungal pathogens invading the mucosal layer [60]. The involvement of ROS systems with microbes occurs in the phagolysosome of the innate immune system, and induces inflammation of tissue damage, especially as it relates to the pathogenesis of autoimmune diseases. Due to metabolic ROS activities, altered apoptosis and autoantigen structure reveal novel epitopes through the unmasking of cryptic determinants to promote the production of autoantibodies and cell immune activation, which initiates the autoimmune disease. In addition, enzyme and biochemical pathways typically produce ROS, which causes metabolic dysfunction and inflammatory regulation. In addition, serum total fatty acids have been explored as a target metabolomic biomarker to predict autoimmune diseases [61]. These fatty acids include saturated fatty acids, such as lauric acid (C12:0), myristic acid (C14:0), stearic acid (C18:0), lignoceric acid (C24:0), and heptadecanoic acid (C17:0); monounsaturated fatty acids, such as palmitic acid (C16:0), Cis-10-pentadecanoic acid (C15:1), Cis-11-eicosenoic acid (C20:1n9), and erucic acid (C22:1n9); and the polyunsaturated fatty acid gamma-linolenic acid (C18:3n6). Quantifying these metabolites permits the identification of the cellular metabolic state even prior to disease onset. Through gas chromatography–mass spectrometry (GC–MS) techniques, previous studies have shown significant metabolism differences during the development of the disease [62]. Findings from studies of randomized clinical trials showed that urinary organic acids, including 3-hydroxypropionic and methylcitric acids, propionylglycine, tiglylglycine, 3-hydroxy-n-valeric, and 3-keto-n-valeric acids, could be potential biomarkers to track the progression of the disease. These biomarkers are also detected in chronic malnutrition leading to imbalance between pro-inflammatory T cells and the regulatory T cells that control inflammation. 

The metabolic switch during T cell activation to control autoimmunity requires the activation of the nuclear receptor 77 (Nur77) as a molecular brake [63]. The role of Nur 77 metabolic processes during T cell activation is to control oxidative phosphorylation and aerobic glycolysis during T cell activation, restricting inflammation in autoimmune conditions. The basis of immunometabolism is associated with immune cell metabolism and control of a complex metabolic network for cell activation and expansion. The complex system is dependent on T cell receptor self-activation in the development of T-cell-mediated autoimmunity, rather than co-stimulatory signals or polarizing cytokines. The analysis of the metabolic transcriptional network via RNA-seq and real-time RT-qPCR revealed that metabolic pathways in autoimmune diseases are under the control of relevant transcription factors, such as Nr2f1, Nr0b1, Esrra, Esrrb, and Essrg, which regulate T cell metabolism and subsequent pro-inflammatory T cell function under modulation of estrogen-related receptor alpha [63,64]. Thus, metabolic gene expression and glucose metabolism of effector T cells are viable approaches for treating T-cell-mediated autoimmune diseases.

## 3. Non-Invasively Accessible Resources for Biomarkers

### 3.1. Sweat

Sweat allows noninvasive sampling and detection, which are crucial for continuous biomonitoring, particularly in neonates and the elderly population. In addition, sweat sample preparation takes minimal time and effort due to the absence of most of the impurities that are otherwise present in blood samples [65]. Moreover, sweat samples can be taken frequently and stored for an extended period, facilitating efficient post-processing. In particular, sweat contains a broad range of analytes, including electrolytes, proteins, and lipids. Yu et al. detected more than 800 unique proteins and 32,000 endogenous peptides in sweat, thereby demonstrating the significant potential of sweat-based diagnostics and prognostics [66]. Notably, sweat hosts a wide variety of disease-specific biomarkers and drug metabolites. Marques-Deak et al. and Dai et al. performed extensive cytokine profiling of sweat samples and identified IL-1α, IL-1β, IL-6, IL-8, TNF-α, and transforming growth factor-beta (TGF-β). These results were verified in a more comprehensive study where 42 human cytokines were identified in 30 sweat samples [65]. This study identified additional cytokines, including IL-2, IL-10, IL-13, Interferon-gamma (IFN-γ), and MCP-1. The cytokine panel included the key adipokines associated with MS: MCP-1, IL-1, IL-6, IL-10, and TNF-α. Sweat is also rich in glucose and uric acid, which are significantly elevated in MS patients with a high cardiovascular risk profile [3,67].

Despite the well-documented list of biomarkers in sweat and the advantage of its noninvasive collection, its unique secretion mechanism, as well as poor collection methods, separate collection and analysis stages, to the impossibility of monitoring multiple analytes simultaneously, and a lack of correlation studies between sweat and blood measurements still hinder the successful clinical translation of sweat-based biosensors [68,69]. Sweat is secreted by the eccrine glands and carries physiologically relevant analytes to the skin surface. This transport mechanism creates a time lag between protein expression in blood and sweat, hindering instantaneous monitoring. Moreover, analyte transport to sweat is also subjected to tight protein junctions along their path [68,69]. These junctions act as barriers to larger biomarkers and dilute them along the paracellular path. For instance, sweat glucose is transported to the skin surface through the paracellular route, and it is 100 times more diluted than interstitial fluid or blood plasma glucose. Extensive dilution of analyte reinforces the need to develop ultrasensitive and highly selective sensors with carefully designed sweat sampling methods. Additionally, much is yet to be understood regarding the correlation of sweat biomarkers to their blood counterparts and the physiological state. This is evident in the existing sweat-based sensors that only monitor fitness indicators, such as sodium and potassium ion concentrations and rate of sweat loss. These sensors do not quantify specific diagnostic biomarkers. 

Nevertheless, efforts are underway to tap the potential of protein-rich and easily accessible sweat for health monitoring. A breakthrough was achieved through the development of integrated sensors for diagnosing cystic fibrosis and benchmarking with existing gold standard laboratory-based bioassays [70,71,72]. Research has also demonstrated the utility of sweat analytes in gout and metabolic disorder monitoring. For instance, researchers developed a laser-engraved wearable sensor that can detect uric acid and tyrosine in sweat [73]. Human studies were conducted to evaluate the performance of the sensor in gout management.

The reliability of individual sweat sensors can be improved by incorporating multiplexed (multiple analyte detection on a single sensing platform) and multimodal (multiple transducers for the same set of analytes) detection. An ideal integrated platform would comprise sweat collection, the transducer unit, signal processing, data transmission, and post-processing. More importantly, the correlation of analyte concentrations in sweat with the pathology of MS needs to be established and clinically validated.

### 3.2. Tears

Tears are secreted by the lachrymal gland and are an excellent source of proteins, lipids, hormones, electrolytes, nucleic acids, and glucose [74]. Biomarkers in tears directly diffuse from the blood through the blood–tear barrier. Hence, the biomarker concentrations in tears are more closely correlated to blood than other biofluids, such as sweat. Basal tears that act as the protective film covering the eye surface continuously contact the blood through the blood–tear diffusion barrier. This blood–tear interaction makes the basal tear an attractive target for noninvasive, continuous, and long-term disease monitoring. 

Several proteins in tear fluid have been found to hold clinical utility in diagnosing ocular diseases, including dry eye syndrome, trachoma, glaucoma, keratoconus, and thyroid-associated orbitopathy, as well as systemic disorders including diabetes mellitus, cancer, systemic or multiple sclerosis, cystic fibrosis, and Alzheimer’s disease. Glucose and lactate are two well-established tear analytes [75,76,77,78,79]. In addition, tear fluid is a rich source for metabolic phenotyping [80]. Tears of diabetic patients show elevated levels of several metabolites, such as carnitine, tyrosine, uric acid, and valine [80]. Increased levels of inflammatory cytokines and chemokines in tears, including IL-6, IL-1, IL-8, IL-10, and TMF-α, have been investigated in patients with dry eye disease. Moreover, drugs such as metformin, bisoprolol, and gabapentin are also detectable in tears [80], thus highlighting the potential of tear fluids in monitoring therapeutic outcomes. However, there is a gap in knowledge regarding the levels of MS biomarkers in tears, and the relationship between tear and blood biomarker levels remains unclear. 

Currently, challenges persist with tear-based biomarker monitoring. For instance, the reflex tears generated during emotional or mechanical stimulation interfere with the basal tear-based sensing methodology, owing to the differences in composition and secretion rates of basal and reflex tears. Other potential challenges include the limited volume of basal tears (less than 5 µL) [81] produced by our eyes, and the need for a biocompatible, stable, energy-efficient, and miniaturized sensing platform with sufficiently high sensitivity and selectivity. These hurdles can be overcome through device miniaturization and biocompatible electronic interfaces.

### 3.3. Breath

Analysis of exhaled human breath allows noninvasive, fast, and personalized detection of health parameters. The volatile organic compounds (VOCs) present in the exhaled breath carry the body’s fingerprints of metabolic and biophysical processes [82]. The VOCs are formed metabolically, brought to the lungs via the bloodstream, and finally exhaled via the respiratory tract [83]. Research has demonstrated the alterations in the breath composition of diabetic patients. Metabolic disorders in diabetic patients, such as increased glucose and lipolysis levels, lead to elevated acetone concentrations in the exhaled breath. Hence, acetone is considered a breath biomarker for identifying impaired metabolism in diabetic patients. Breath acetone level ranges from 0.3 to 0.9 ppm in healthy individuals, whereas the levels can reach up to 1.8 ppm in diabetic patients [84,85]. Diabetic patients also exhale a higher concentration of isopropanol than a healthy individual [86]. Individuals suffering from a cluster of metabolic disorders would exhale a different concentration of acetone and isopropanol. Extensive investigations are required to dissect the breath signatures related to MS and unravel its underlying pathogenesis. 

The existing breath analysis is centered around mass spectrometry owing to its ppb-level sensitivity [87]. Although mass spectrometers provide ultra-high sensitivity, they entail time-intensive and disruptive measurements, lack portability, and prevent follow-up and dynamic studies. Hence, mass-spectrometry-based techniques are difficult to implement on a wide scale. Conversely, a common challenge encountered by commercial portable electronic nose technology is the lack of sensitivity to VOCs present at exceptionally low levels (i.e., at the ppb level and below) in the exhaled breath [83]. With rapid advances in nanotechnology and molecular diagnostics, the use of breath for diagnosis is evolving. The emerging detection methods should take the aforementioned limitations into account and exhibit high sensitivity and selectivity to the breath analytes for noninvasive MS screening. 

### 3.4. Saliva

Saliva is an attractive biofluid for quantifying biomarkers related to oral and systemic diseases due to the noninvasive nature of saliva testing and its continuous availability [88]. Several biomarkers have been detected in saliva, including alpha-amylase, glucose, lactate, phosphate, and uric acid. Their salivary levels correlate well with their blood levels [89]. Research demonstrates the clinical utility of salivary biomarkers in the identification of MS. Salivary C-reactive protein levels exhibit consistent results in type 2 diabetes, while salivary adiponectin, leptin, IL-6, and TNF-α provide inconsistent results, requiring additional investigations [90]. 

Researchers have developed mouthguard and pacifier sensors for continuous monitoring of metabolites (including uric acid, lactate, and glucose) from saliva [91,92,93]. The electrochemistry-based technique was used to transduce metabolite levels into measurable electrical signals. Additionally, smartphone-based portable electrochemical biosensing system for salivary microRNAs has been introduced for point-of-care in remote areas [94]. Some challenges remain in translating these saliva sensors into clinical practice. These challenges include a lack of continuous and reliable operation of these devices over a long period, and their insufficient sensitivity to ultra-low concentrations of analytes found in saliva (sometimes even lower than those found in sweat) [88].

### 3.5. Urine

Urine is another rich source of metabolites, as it contains salts, proteins, and other clinically useful analytes. Compared to other body fluids, urine samples can be obtained in large quantities, which can be repeated several times a day. Molecules diffuse into urine from nearby blood capillaries; hence, their urinary levels correlate well with their blood counterparts. Several key parameters are identified in urine, including glucose, ketones, bilirubin, and bacteria. Abnormal urinary levels of these parameters may indicate diabetes, kidney diseases, renal diseases, bacterial infections, dehydration, and bladder cancer [95]. Although metabolomic analysis of urine is a mature process, translating laboratory-based assays to the bedside is a challenge. Nevertheless, intelligent assays, such as biomarker harvesting hydrogel nanoparticles and affinity-capture pre-processing techniques, are enabling the quantification of previously undetectable urinary proteins [96].

## 4. Biosensor Platforms

### 4.1. Electrochemical Biosensors

Chemical detection is a straightforward method of monitoring targeted molecules, including biomarkers for MS. Owing to the nature of chemical reactions, the sensors exhibit high specificity. Among them, electrochemical sensors have been widely investigated due to their high sensitivity and miniaturization advantages. Electrochemical sensors present high temporal and spatial resolution due to their micro-size and response time in seconds [97,98]. Furthermore, their high reaction speeds enable real-time and continuous monitoring of target molecules. Furthermore, batch production of probes is possible with the commercial clean room fabrication process, bringing the total cost of the sensor down to an affordable range. Recently, with the advance in technology, 3D-printed electrochemical sensors have been introduced into the field to compensate for the time-intensive labor required for the existing clean room fabrication process, making the fabrication process easier and more cost-effective [99].

Because electrochemical sensors operate based on the electroactivity of target molecules, they can be classified into two categories: sensors targeting electroactive molecules and those targeting non-electroactive molecules. In the case of electroactive molecules, the fabrication process is relatively simple, as the sensing material and optimized potential are the only relevant factors in optimizing the reaction occurring on the sensor surface. Examples of these molecules are vitamin C (ascorbic acid), dopamine, and uric acid [100,101,102]. However, for molecules that are not electroactive, additional mediators are needed to convert these non-electroactive molecules into a reporter molecule that can be either reduced or oxidized at the sensing electrode. These molecules include glucose, lactate, L-glutamate, GABA, etc. [103,104,105]. As MS is a metabolic disorder that results from the imbalance of several molecules, various electrochemical sensors have been reported to monitor the analytes that are related to MS, as discussed in the previous section.

#### 4.1.1. Cardiovascular Disease

As explained earlier, aside from behavioral patterns, various biomarkers play a role in developing CVD. Monitoring these molecules can contribute to the early diagnosis of CVD, possibly increasing the survival rate for patients with the disorder. In this regard, Boonkaew et al. developed a multiplexed electrochemical paper-based analytical device (ePAD) for simultaneous detection of three CVD biomarkers, CRP, troponin I (cTnI), and procalcitonin (PCT) (Figure 4a) [106]. The antibodies for each target analyte were immobilized on the graphene oxide (GO)-modified carbon electrodes printed on the ePAD. The change in current due to the binding of target biomarkers was observed with square wave voltammetry, and the current was linearly correlated to the analyte concentration. With the advantage of using a graphene-modified stencil-printed carbon electrode, the sensor showed a wide detection range of 1–1000 ng mL^−1^, 0.001–250 ng mL^−1^, and 0.0005–250 ng mL^−1^ and a limit of detection of 0.38 ng mL^−1^, 0.16 pg mL^−1^, and 0.27 pg mL^−1^ for CRP, cTnI, and PCT, respectively, which is sufficient when compared to the values suggested by the WHO.

Koukouviti et al. reported an alternative type of multiplexed electrochemical sensor for monitoring cardiac biomarkers (Figure 4b) [107]. A 3D-printed 4-electrode enzymatic biosensor for simultaneous measurement of cholesterol and choline was developed by this group. Notably, the overall cost for the biosensor was brought down to $0.031, and the fabrication time was reduced to around 30 min by 3D printing of the whole chip. This eliminates the time-consuming process of conventional biodevice fabrication, which includes sputtering and photolithography. Additionally, it allows on-demand manufacturing at the time of need. As a proof of concept, simultaneous amperometric measurement of two cardiac biomarkers in the blood droplet, cholesterol and choline, was performed, and the LODs were found to be 3.36 and 0.08 μmol L^−1^ for cholesterol and choline, respectively, which are much lower than the threshold levels for coronary syndromes (6 mmol L^−1^ and 28 μmol L^−1^ for cholesterol and choline, respectively).

#### 4.1.2. Prediabetes

Recently, research regarding multiplexed electrochemical measurement of glucose has been guided towards integrating various measuring techniques into a single platform.

Rogers et al. reported a wearable microfluidic system that can perform flow immunoassays, fluorometric assays, and digital galvanic measurements to track the stress states [108]. Various reporters for stress have been monitored through the platform, including cortisol, vitamin C (ascorbic acid), glucose, sweat rate, and galvanic skin response. First, the lateral flow immunoassays (LFIAs) targeted cortisol in sweat using anti-cortisol antibodies (ACA) and gold nanoparticles (AuNPs). The hydrophobic surface of AuNPs assisted in the conjugation of ACA and AuNPs, which reacted with cortisol in sweat to generate a signal. The cortisol-ACA-AuNP bound to the anti-mouse IgG (anti-IgG) antibody in the test channel, where the signal was correlated with the number of AuNPs, the number of binding sites per AuNP, and the total amount of cortisol-BSA, and the concentration of sweat cortisol was measured.

Similarly, Javey et al. reported a fully integrated wearable sensor array for multiplexed sweat analysis made by integrating skin-conformal plastic-based sensors and commercially available integrated circuit components [111]. This platform measures glucose, lactate, potassium, sodium, and skin temperature sensitively and selectively. The utilization of flexible polyethylene terephthalate (PET) enabled stable sensor–skin contact for better surface area with higher sensitivity. For glucose and lactate detection, corresponding enzymes were functionalized on the electrode surface to convert the analytes into electroactive reporters. For ion detection, ion-selective electrodes (ISEs) were used. Additionally, metal microwires were used to monitor the temperature based on its resistance. The performance of the sensors was found to be aligned with the physiological concentrations, demonstrating their health-monitoring application not only for the general public but also for athletes whose loss of ions can crucially affect their bodily function. Furthermore, this platform can be reconfigured for in situ analysis of other biomarkers for real-time physiological and clinical investigations.

Although the original purpose of these multiplexed sensors was for general health monitoring and not for diagnosing MS, the target molecules, including ascorbic acid, glucose, lactate, etc., are good indicators for MS. Thus, by repurposing its use from general health monitoring to specific MS diagnosis via modification of its components, this multiplexed, multimodal sensing system can overcome the limitations of single analyte sensors and provide a highly sensitive and precise diagnosis platform. The breakthrough made by the aforementioned research groups takes a step further from sensing only the individual’s physical activities and vital signs to a molecular level sensing of the user’s health. Similar to the incorporation of heart-rate monitoring in smart devices, vital-sign measurements are widely used due to their noninvasive nature. However, due to the limitations in access to samples, the biomolecular assay for personalized health has always been challenging.

#### 4.1.3. Cancer

Cancer has recently gained recognition as a metabolic disease. In the past, it has been purely classified as a disorder of proliferation, but recent studies suggest that the metabolic system becomes affected by the overgrowth of tumor cells, leading to a feedback loop of continuous cell growth. Xu et al. reported a multiplexed electrochemical biosensor platform to detect various prostate cancer biomarkers (Figure 4c) [110]. This platform adapted bioinspired superwettability, which combines two extreme states of super-hydrophobicity and super-hydrophilicity to control the behaviors of liquid droplets. By applying this technique to electrochemical sensing, the amount of solution could be decreased from a typical large volume to a single microdroplet. With the super-hydrophilic surface being the actual sensing area and the superhydrophobic surface surrounding it, a microwell for the microdroplets was formed. In addition, a gold nanodendritic structure was electrode-posited on the surface to increase the adhesion with water microdroplets. The sensor was tested for three different prostate cancer biomarkers, miRNA-375, miRNA-141, and prostate-specific antigen, and it showed sensitive and selective detection properties. 

Another multiplexed platform that was reported for prostate cancer biomarkers was from Tang et al. This group adapted an array of 32 electrochemical sensors to detect four different biomarkers for prostate cancer, which were prostate-specific antigen, prostate-specific membrane antigen, interleukin-6 (IL-6), and platelet factor-4 (PF-4) (Figure 4d). Relative antibodies were coated on the designated sensing sites with eight sensors for each target. Among those eight sensors, two sensors were selected to serve as negative monitors for any nonspecific protein absorption on the sensor surface. The results showed that the platform obtained clinically relevant detection limits (0.05 to 2 pg mL^−1^) and 5-decade dynamic ranges (sub pg mL^−1^ to well above ng mL^−1^), indicating its application not only in the early detection of prostate cancer but also in diseases that require the simultaneous detection of several biomarkers.

#### 4.1.4. Autoimmune Disease

In autoimmune disease, where the body’s immune system malfunctions and does not distinguish between the patient’s own cells and foreign cells, the overactivity of the immune system attacks the normal cells in addition to the foreign cells, causing detrimental effects on the patient. There are over 80 autoimmune diseases that have been reported, including rheumatoid arthritis (RA), type 1 diabetes, systemic lupus erythematosus (lupus), and inflammatory bowel disease, among others. Currently, treatments for autoimmune disease primarily focus on treating the patient once the condition occurs, where the focus is on reducing the activity of the immune system. However, it would greatly benefit patients if early diagnosis were possible before any adverse and irreversible damage occurred.

RA is one chronic autoimmune disease that occurs in the joints of the hands and feet. While the exact cause of the disease is still unknown, the production of IgM rheumatoid factor (RF) and anti-cyclic citrullinated peptide/protein antibodies (anti-CCP-ab) are considered to cause RA. Considering that a vast amount of people with positive anti-CCP-ab have a 5-year risk of developing RA, early detection of anti-CCP-ab can significantly help people recognize RA before the disease enters the chronic phase. In this regard, Cho et al. have reported an electrochemical sensor designed to detect anti-CCP-ab for early detection of RA [112]. The acceptable level of anti-CCP-ab in healthy individuals is below 20 IU/mL, so levels above this number can indicate the subject’s susceptibility to developing RA. Using an avidin-biotin bio-recognition system and self-assembled monolayer (SAM) of mercaptohexanoic acid (MHA), the authors achieved a platform that can maintain the activity for the immunogen used in this study, B-CCP, with good selectivity. Furthermore, the sensing platform was compact (14 × 3.5 mm) due to the MEMS structure, with remarkable functionality for performing electrochemical impedance spectroscopy (EIS). As the concentration of anti-CCP-ab changes in human serum (HS), the charge transfer resistance of the EIS measurement changed accordingly. This sensor showed good linearity from 1 IU/mL–800 IU/mL with a limit of detection of 0.82 IU/mL in HS. In addition, the sensor was able to function for over 20 days with good stability, making it suitable for point-of-care (POC) devices.

Table 2 outlines some electrochemical sensors reported in the literature for multiplexed detection of MS biomarkers. 

### 4.2. Optical Biosensors

Numerous studies have been conducted to develop optical sensors for single analyte detection. Figure 5 depicts an overview of different types of optical sensing platforms, including meta-structures, surface plasmon resonance (SPR), reflectometric interference [126], evanescent wave fluorescence [127], bioluminescence [128], and surface-enhanced Raman scattering (SERS) [129]. However, little progress has been made on multi-analyte detection using a single POC framework. MS is a constellation of multiple disorders, thereby necessitating multiplexed monitoring with a single sensing platform. Sensors that are cost-effective and capable of monitoring the body’s initial responses to metabolic disorders, such as abnormalities in the immune system response, adipokines, and uric acid levels, as well as the resulting dysfunctions, such as elevated levels of blood glucose, triglycerides, cholesterol, oxidative stress, LDL, and insulin resistance, hold substantial potential for affordable and large-scale screening of populations. In addition, the early-detection capabilities of these sensors would reduce mortality and improve quality of life.

#### 4.2.1. Multiplexed Optical Detection Systems

Multiplexed monitoring is crucial because MS involves an interplay between several biological processes. Expression of one biomarker can indicate more than one disease, different disease conditions may manifest with identical physical symptoms, and it may be paramount to monitor the dynamic progression of a disease [130]. In addition, multiplexing generates high-throughput spatiotemporal data while conducting multiple tests simultaneously. Subsequently, large sets of data can be combined with an artificial intelligence/machine learning (AI/ML) framework to implement just-in-time interventions, predict the progression of diseases, and improve treatment outcomes. This section first describes the conventional techniques of multiplexed biomarker detection and then transitions to point-of-care (POC) and in vivo detection schemes. 

The current gold standard methods for multiplexed detection of molecular biomarkers include enzyme-linked immunosorbent assay (ELISA), fluorescence immunoassays, and polymerase chain reaction (PCR). ELISA employs multiple wells to probe distinct analytes. Receptor–analyte interaction inside each well generates a colored product, the optical density of which is proportional to the analyte concentration [131]. Although the multiple-well framework enables biomarker multiplexing in a single assay, ELISA has certain limitations, including lack of standardization among similar assays, laborious assay procedure, expensive reagents and plate reader, specialized storage and handling, and often, the need for higher sample volume (>100–200 µL) [131,132]. In contrast to enzymes used with ELISA, fluorescence immunoassays use a fluorescent label for quantifying proteins [133]. Fluorescence assays are also unsuitable for POC applications due to additional limitations, such as low photostability and intrinsic background fluorescence [134]. PCR is the current gold standard for the qualitative identification and differentiation of infectious species, by detecting nucleic acid sequences unique for the pathogen. PCR can identify different types of organisms, including bacteria, parasites, viruses, and fungi, from their genetic signature. Although the working principle and ingredients are similar, to ensure selective detection of the pathogen and reduce false-positive results, specific primers and probes are used to detect different organisms. In addition, PCR only works on DNA, hence, prior to PCR, the viral RNA is converted to DNA using a reverse transcription (RT) mechanism and the PCR is referred to as RT-PCR [135]. While molecular tests are highly sensitive and widely used for molecular biomarkers analysis, running the tests and analyzing the results can take up to a week (in locations with many tests) and require sophisticated lab equipment and technicians. Moreover, molecular tests cannot monitor disease progression or past infection in real time.

Recent progress has been made on screening multiple biomarkers at the POC. Some notable architectures are explained hereafter. Jahns et al. reported a photonic crystal biosensor for multiplexed detection of CD40 ligand antibody, EGF antibody, and streptavidin in parallel on a single chip [136]. The photonic crystal comprised titanium-dioxide-coated periodic grated nanostructures functionalized with multiple specific ligands, as depicted in Figure 5 (left panel). A simple CMOS camera and imaging optics were used to record and analyze the spatially resolved transmission intensity, thereby eliminating the need for complex optics and spectrometer. The optical transmission spectra exhibited shifts in the guided mode resonance wavelength in response to analyte binding to the receptors, resulting from refractive index changes on the photonic crystal surface. This shift in spectra is schematically shown in Figure 5 (right panel). The green color channel of the camera was read out to measure the intensity shifts in the transmission spectrum. Figure 5 (middle panel) demonstrates the experimental measurement setup in which the multiplexed photonic crystal biosensor was placed between two crossed polarization filters, and variations in the transmission intensity were recorded by the CMOS image sensor. Another group introduced a novel common-path interferometer biosensor for simultaneous sampling of tens of measurement fields [137]. The system utilized multi-pinhole Fourier frequency division method to measure proteins binding to a photonic crystal surface. A CMOS camera was used to obtain the far-field interference pattern. Figure 5 shows the pinhole plate and Fourier lens-based measurement setup and the shifts in phase and reflection intensity of guided mode resonance upon changes in refractive index at the photonic crystal surface. The pinhole patterns were designed in such a way that their positions matched the different biomarker binding sites. The pinhole patterns in turn determined the frequency positions in the Fourier domain. To increase the number of measurement sites, a spiral pattern was designed with 54 pinholes and three different radii. A silicon photonic microring resonator array was also developed for ultrasensitive detection of four microRNA targets present in standard solutions and extracted from mouse brain tissue [138]. The microring resonator array was functionalized with antibodies (named S9.6) that specifically recognized DNA:RNA heteroduplexes. Figure 5 illustrates the sequential formation of the S9.6 amplification assay and the resulting relative shift in resonance for seven different concentrations of each target microRNA. Chen et al. developed localized surface plasmon resonance (LSPR) microarrays with 480 nanoplasmonic sensing spots in microfluidic channel arrays for multiplexed detection of six serum cytokines in infants following cardiopulmonary bypass (CPB) surgery [139]. The biochip was composed of eight parallel microfluidic channels running orthogonally to six antibody-functionalized gold nanorods with 10 turns on a glass substrate (Figure 5), thereby giving rise to an array of 480 LSPR biosensing spots on the entire chip. The ensemble of around 2000 plasmonically uncoupled gold nanorods on each sensor spot exhibited a distinct resonance, which was leveraged to quantitatively identify multiple cytokines in parallel. Optical fiber opens a new paradigm in biosensing, owing to its smaller footprint, ease of multiplexing, remote monitoring capability, and immunity to electromagnetic interference. A multiparametric fiber-optic probe was realized for simultaneous detection of temperature, curvature and strain [140]. The sensing probe was made of a Bragg grating structure engraved in a graded-index multimode fiber and a Fizeau cavity. All these components were made in series along the fiber. The sensor structure and the experimental setup are shown in Figure 5. This sensor probe can be reconfigured for monitoring biomolecules. Qu et al. reported a duplex fiber-optic LSPR probe functionalized with His_6_-tagged T2C2 and MDTCS bioreceptors to detect the two antibodies, 4B9 and II-1 [141]. Gold nanoparticles served the dual purpose of signal amplification and distinguishing the two targets in the same sample. The same fiber was sequentially functionalized with the bioreceptors, expanding its applications in several biosensing fields. Other noteworthy multiplexed protein monitoring platforms include spotted arrays of randomly distributed metal nanoparticles functionalized with specific targets that generate graded SPR [142], Raman dye-coded aptamer-gold nanoparticle conjugates [143], and plasmonic staining of inverse opal photonic crystal hydrogel bead [144]. Although these multiplexed frameworks provide POC monitoring, they are particularly suitable for formulating *in-vitro* protein assays. Challenges persist in the realization of in vivo multiplexed optical sensors.

Recent advances in fiber optics have led to the realization of optofluidic and optoelectronic platforms, which can be reconfigured for multiplexed in vivo applications. Frequently, the fiber structures are micromachined to produce sophisticated core-and-shell geometry for integrated optofluidic analysis [145]. Such architectures involve sophisticated lithography and functional structure integration processes. Other approaches include the incorporation of functional materials into microstructured fibers to realize multimodal and flexible neural probes. This is particularly applicable to the spinal cord and the peripheral nerves, which are subject to frequent bending and stretching deformations [146]. Some design considerations for developing neural probes include minimizing tissue damage and foreign body response, increasing flexibility of the devices, and adding surface coatings to incorporate biocompatibility. These features are difficult to achieve with conventional neural interfaces owing to their rigid geometry and lack of mechanical compliance. In contrast, optical fibers can be readily maneuvered and programmed with elasticity and stretchability, multiple materials, and micro and nanoscale features to circumvent the hurdles in traditional neurological studies. For example, thermo-resistive optical fibers were fabricated by incorporating novel chalcogenide semiconductor Ge_17_As_23_Se_14_Te_46_ core, and metallic electrodes (Sn_96_Ag_4_) [147]. In addition, novel tunable architectures were reported for chemical sensing, such as a hollow channel surrounded by alternating layers of chalcogenide glass arsenic selenide (As_2_S_3_) and polyetherimide [148], as well as a hybrid all-in-fiber configuration [149]. Stretchable and tunable fibers have also been reported in applications with biological tissue interfaces [150]. Other emerging techniques include incorporating multiple probing [151] and stimulation [152] sites and discrete devices (e.g., micro LEDs and photodetectors) [153] along the fiber.

#### 4.2.2. Cardiovascular Disease

As explained earlier, cardiac troponin, CRP, PCT, and IL-6 biomarkers play a crucial role in developing CVD. Kundu et al. developed a plasmonic POC device comprising a gold- and graphene-oxide-coated patterned array of periodic nanoposts to detect PCT [154]. The sensor demonstrated a sensitivity of 0.0643 a.u./pg·mL^−1^ at lower concentrations and 0.0224 a.u./pg·mL^−1^ at higher concentrations of PCT, and a detection limit of 1.22 pg·mL^−1^. The sensor chip was capable of measuring the kinetics of antigen–antibody bindings at the SPR sensor surface. In addition, the sensor exhibited high reproducibility to PCT molecules down to the picomolar level, which could be attributed to the soft lithography-based low-cost sensor manufacturing process. SPR sensor was also used for multiplexed detection of myocardial infarction biomarkers, cardiac troponin I (cTnI), and myoglobin (MG) [155].

Two highly sensitive fiber-optic biosensing techniques include fluorescence and label-free methods. A combination tapered fiber-optic biosensor was developed by forming a molecular sandwich assay near the tip [156]. The fiber probe was functionalized with capture antibodies and immersed in a solution containing fluorescence-dye-labeled detection antibodies. In the absence of target antigens, the evanescent field of the fiber had few to no interactions with the dye, and hence little fluorescence was observed. Upon addition of the antigen, a molecular sandwich was formed on the probe surface and the evanescent field excited the fluorescent dyes. This combination tapered probe was used to detect IL-6 with a limit of detection of 5 pM. In another work, a fluoro-mediated sandwich immunoassay was made on an optical fiber for simultaneous detection of four CVD biomarkers, B-type natriuretic peptide (BNP), cardiac troponin I (cTnI), C-reactive protein (CRP), and Myoglobin (MG) [157]. Noushin et al. reported dual-modality detection of IL-6 by integrating a fiber-optic sensor into a microfluidic electrochemical sensor on a wearable biochip [158]. Combination of two modalities allowed the detection of a wide range of IL-6 molecules in sweat samples. For instance, the electrochemical sensor had a linear operation range from 0.1 pg/mL to 1000 pg/mL, while the fiber-optic sensor operated from 1 ng/mL to 1000 ng/mL of IL-6. 

The aforementioned literature was not focused on detecting CVD in particular, as some biomarkers are expressed in a wide range of diseases. For instance, CRP is a general inflammatory biomarker. Therefore, more studies need to be conducted to apply the already reported sensors to MS diagnosis.

#### 4.2.3. Prediabetes

Different types of nanomaterials have been incorporated into optical sensing modalities to enhance sensitivity and selectivity to MS biomarkers, such as glucose. For instance, microgels were assembled on an SPR fiber tip to develop a fiber-optic glucose sensor with tunable limit of detection, working range, and response time [159]. The 3D porous network of microgels allowed the detection of millimolar glucose concentrations, previously undetected with a planar platform. The sensor exhibited a linear detection range from 16 μM–16 mM and a tunable limit of detection was achieved by changing the microgel concentrations.

Koman et al. introduced a multiscattering-enhanced optical biosensor for multiplexed, continuous, and non-invasive monitoring of lactate and glucose [160]. The sensing spots were deposited on a porous membrane that demonstrated spatial variations in refractive index leading to multiple scattering of light. Subsequently, an inverted microscope was used to obtain the absorption spectra from the sensing spots under white-light illumination. Although the original purpose of this multiplexed sensor was for continuous measurements of cellular processes, such as monitoring the uptake of exogenously supplied glucose by the green algae *Chlamydomonas reinhardtii*, the sensing platform can be readily repurposed for diagnosing MS.

Kim et al. developed a paper-disc centrifugal optical device for colorimetric detection of glucose and lactate [161]. The sensing platform comprised of a disposable, wax printed, rotting paper module along with a photodiode and light-emitting diode to quantitatively measure the color changes in multiple reaction zones on the paper-disc in real time. Furthermore, the device had a self-calibration feature that minimized the effects of ambient light, temperature, humidity, and measurement time variations. 

#### 4.2.4. Cancer

With the emerging evidence that cancer is a metabolic disease, it is worth investigating the existing sensing technologies that would inform the repurposing of the sensors for diagnosing MS. A variety of high-performance and label-free optical sensors have been reported in the literature for detecting cancer biomarkers. For instance, a fiber-optic LSPR probe was developed by Sanders et al. for label-free detection of prostate-specific antigen (PSA), an important biomarker for prostate cancer [162]. The LSPR signal was excited by a gold nanodisc array made at the end facet of the fiber using electron beam lithography and lift-off techniques. Anti-PSA monoclonal antibody molecules were functionalized at the LSPR probe for selective detection of PSA. Experimental results demonstrated a detection limit of 100 fg/mL. 

Surface-enhanced Raman spectroscopy (SERS) is another powerful tool for ultrasensitive detection of chemical species. SERS involves a highly localized field enhancement induced by laser excitation of roughened metallic surfaces or metal nanostructures. The enhancement factor can reach up to 10^11^, leading to the possibility of single molecule detection [163]. Several SERS biosensors have been investigated for multiplexed detection of prostate cancer biomarkers, including a silver-nanoparticle-based sensor for the detection of PSA, PSMA, hK2 (Human kallikrein 2) [164], core-shell SERS nanotags [165], and gold nanorod SERS nanotags [166] for the detection of PSA, CEA (carcinoembryonic antigen), and AFP (alpha fetoprotein). A more comprehensive list of multiplexed biosensors for prostate cancer diagnosis can be found in [167]. In addition, a chip-integrated silicon photonic sensor array was developed for multiple-cancer biomarker detection [168]. The biochip was used to detect eight different cancer biomarkers: AFP, activated leukocyte cell adhesion molecule (ALCAM; breast cancer biomarker), cancer antigen 15-3 (CA15-3; breast cancer biomarker), cancer antigen 19-9 (CA19-9; pancreatic, colorectal, and ovarian cancer biomarker), cancer antigen-125 (CA-125; ovarian cancer biomarker), CEA, osteopontin (ovarian and liver cancer biomarker), and PSA.

#### 4.2.5. Autoimmune Disease

Several optical biosensing platforms have been developed to detect the biomarkers expressed in autoimmune disease. Some notable ones are highlighted in this section. MiRNAs are considered promising biomarkers for detecting a wide range of diseases, including rheumatoid arthritis (RA). Huang et al. developed a nanophoton switch based on quantum dots (QDs) and graphene oxide (GO) for detecting miRNA-21 and miRNA-155 [169]. The transducing mechanism relied on fluorescence quenching based on fluorescence resonance energy transfer (FRET). A fluorescence quenching effect took place between the QDs and GO, and was used as a fluorescence switch with a high signal-to-noise ratio (SNR). The same pair of miRNAs were also detected by an ultrasensitive SPR sensor based on two-dimensional antimonene nanomaterial and Au nanorods [170]. The sensor demonstrated a detection limit down to 10 aM, one of the lowest reported to date. This ultrasensitivity could be attributed to the substantially stronger interaction of antimonene with ssDNA than graphene. Other significant optical biosensors that will serve as a promising avenue for early diagnosis of RA include multiplexed miRNA detection via enzymatic signal amplification [171] and label-free miRNA detection using arrays of microring resonators. The multiplexed sensor presented in [171] detected six different miRNAs (miRNA-21, miRNA-26a, miRNA-29a, miRNA-106a, miRNA-222, and miRNA-335), while the sensor in [172] detected miRNA-21, miRNA-24, miRNA-133b, and let-7c-5p. 

Table 3 outlines some optical sensors reported in the literature for multiplexed detection of MS biomarkers. 

## 5. Outlook: Towards Multimodal Sensor Platforms

The discussion above suggests that combining sensing methodologies and multiplexing may be a more suitable approach, depending on the desired application and type of analysis. With the appropriate blend of sensing modalities, it would be possible to achieve higher sensitivity and selectivity, as well as a wider dynamic detection range, which is beyond the capability of a single sensing mechanism. In addition, fusing multiple-sensing modalities on a single biochip offers improved detection reliability. Electrochemical and optical sensing have their own unique advantages that can be exploited in a single platform by combining the two modalities [173,174]. Although electrochemical sensors are relatively simple to use and easy to miniaturize, several instances, such as real-time monitoring of critically ill patients in an ICU (intensive care unit) setting, require the use of minimally invasive sensors that can be inserted inside the body with ease. Research demonstrates the development of implantable electrochemical sensors for in vivo monitoring [175,176,177,178]. However, such in vivo detection strategies require invasive surgery, which is often not feasible for neonates, seniors, or critically ill patients. In this regard, the unique features of optical fiber enable the realization of multifunctional and multi-responsive in vivo sensing probes [126]. Placement of fiberoptic catheters into the internal jugular veins has been demonstrated to provide continuous venous oxygen saturation measurements. However, challenges persist in the realization of multiplexed fiber-optic biosensing probes. Conventional optical fiber-based multiplexed biosensing involves a bundle of optical fibers, which substantially increases the footprint and the number of coupling elements between the fibers and the free space [179,180]. In contrast to these setups, a fiber-optic device that does not require any cleanroom procedures or costly physical operations (e.g., etching, surface cleaning, and surface preparation), is highly desirable for low-cost manufacturing of sensor systems. In addition, it can provide multiplexed monitoring with a smaller footprint and offers functionalities comparable to conventional devices, with tremendous scientific and technological merit and clinical applications. Moreover, optical sensors are not affected by interferences, such as magnetic, ionic, or electric fields [173]. In contrast, electrochemical sensors do not require a dark environment for measurements and have a wider dynamic range compared to optical sensors [173]. Furthermore, biosensors with clustered, regularly interspaced, short palindromic repeats (CRISPR)/Cas effector is widely being studied recently due to its excellent properties [181]. One example is using collateral cleavage activity from Cas13a, where the CRISPR/Cas system cut labelled RNA reporters for signal generation at room temperature, once bound to the target gene [182]. In this regard, this novel technique has been integrated with electrochemistry with higher sensitivity and accuracy [183]. Hence, adapting the strengths from electrochemistry and optics and combining these two modalities will generate a richer set of data than would be possible with a single modality. Moreover, the vast quantities of data generated from the multimodal and multiplexed platforms can further be analyzed using meta-analysis to determine precise biomarkers for the diagnosis of MS [184]. As MS is a complex and multi-factorial condition, computational methods can provide information on the difference in the level of importance for each biomarker, and contribute to forming personalized treatments for effective patient care [185,186,187].

The improvement in device performance was also verified in our previous work, wherein we developed dual-modality transducing units (optical and electrochemical) on a single integrated platform for monitoring protein biomarkers in a microfluidics [188] and flexible wearable setting [158]. The two sensing modalities were demonstrated to complement each other in terms of dynamic operation range and detection capabilities [158]. Another work regarding the integration of modalities was reported by Lamberti et al., where optoelectrochemical detection of insulin was performed on a graphene-modified substrate [189]. This sensor showed improved sensing reliability and efficiency, including redundancy of detection, internal titration, efficiency verification properties, and avoidance of false positives. Therefore, in the diagnosis of MS, for which dynamic interactions and interrelations among risk factors need to be considered, multimodal and multiplexed sensing platforms integrating electrochemistry with optics can further enhance the overall performance of the biosensor platform and provide more accurate results.

## Figures and Tables

**Figure 1 sensors-22-05200-f001:**
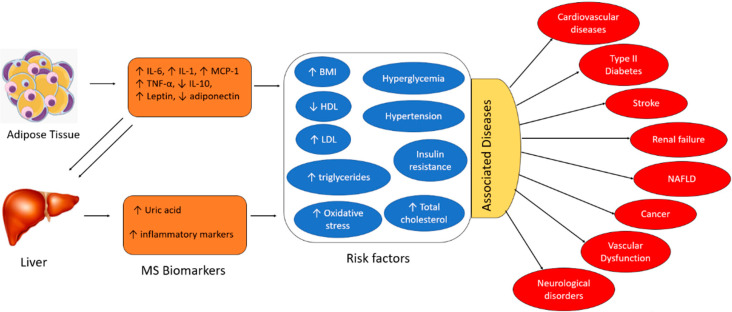
Interaction of adipokines and inflammatory biomarkers that contribute to the development of metabolic syndrome and its associated risk factors and diseases (HDL: high-density lipoprotein, LDL: low-density lipoprotein, NAFLD: non-alcoholic fatty liver disease, BMI: body mass index).

**Figure 2 sensors-22-05200-f002:**
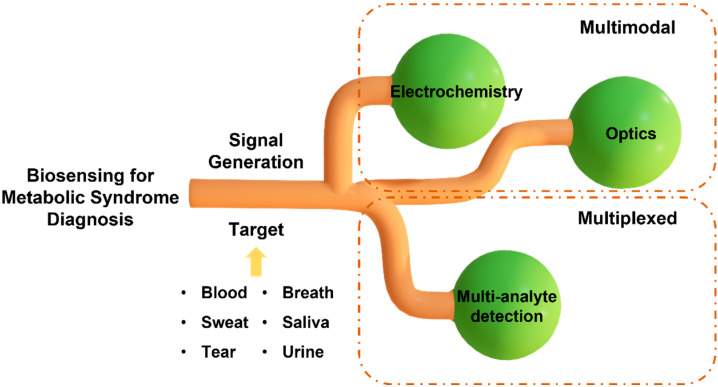
Schematic illustration of multiplexed and multimodal biosensing for diagnosis of metabolic syndrome.

**Figure 3 sensors-22-05200-f003:**
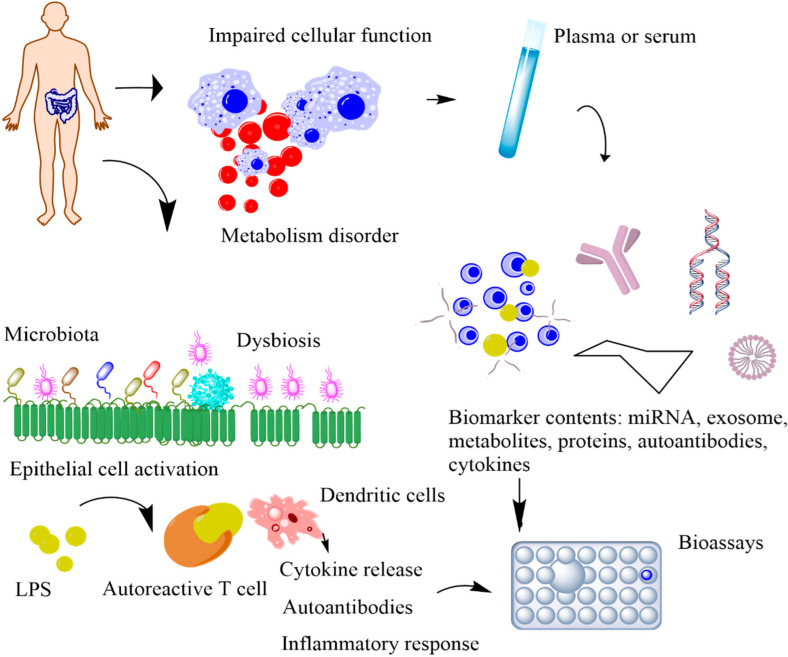
Schematic representation of the metabolic biomarkers released by impaired cellular functions and immune-gut microbiota interactions. In impaired cellular function, released biomarkers include exome, miRNA, cellular components, and antibodies. In immune–microbiome interactions, the intestinal dysbiosis and the increased bacterial lipopolysaccharides (LPS) trigger the autoimmune response, causing disease onset.

**Figure 4 sensors-22-05200-f004:**
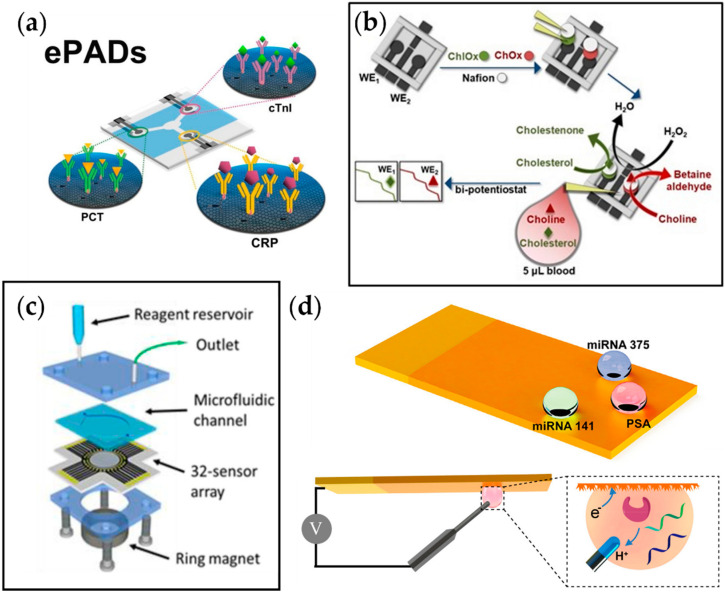
Examples of multiplexed electrochemical sensors for metabolic diseases. (**a**) Multiplexed ePAD for cTnI, PCT, and CRP detection [106]. (**b**) Fabrication process of the 3D printed biosensor for amperometric detection of cholesterol and choline [107]. (**c**) A 32-sensor array with microwells with a microfluidic chamber for simultaneous detection of prostate cancer biomarker proteins [109]. (**d**) Fabrication process of superwettable electrochemical microchip and schematic of electrochemical detection of analyte in droplets formed on the sensor [110].

**Figure 5 sensors-22-05200-f005:**
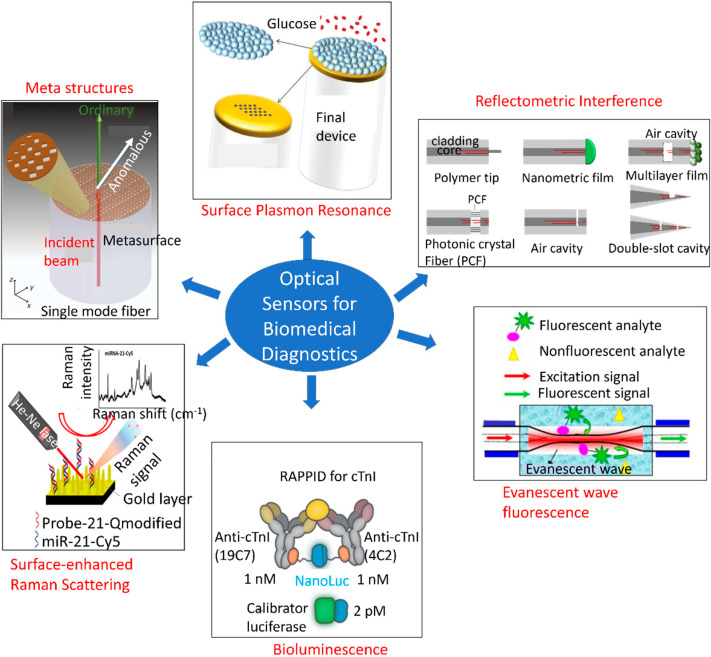
Overview of different types of optical sensing mechanisms based on the optical phenomenon arising from receptor–analyte interactions. This overview shows meta-structures, surface plasmon resonance, reflectometric interference [126], evanescent wave fluorescence [127], bioluminescence [128], and surface-enhanced Raman scattering (SERS) [129]. Reproduced under the terms and conditions of the Creative Commons CC BY license.

**Table 2 sensors-22-05200-t002:** Electrochemical sensors for multiplexed detection of biomarkers for metabolic syndrome. Abbreviations: SLE, systemic lupus erythematosus; PSA, prostate-specific antigen; PSMA, prostate-specific membrane antigen; BAFF, B-cell activation factor; APRIL, a proliferation-induced signal; SWV, square wave voltammetry; EIS, electrochemical impedance spectroscopy; ECL, electrochemiluminescent; DPV, differential pulse voltammetry.

Metabolic Syndrome	Biomarker	E-Chem Method	LOD	Linear Range	Refs.
Cardiovascular Diseases (CVDs)	C-reactive protein (CRP)	SWV	0.38 ng mL^−1^	1–10,000 ng mL^−1^	[106]
	Troponin (cTnI)		0.16 pg mL^−1^	0.001–250 ng mL^−1^	
	Procalcitonin (PCT)		0.27 pg mL^−1^	0.0005–250 ng mL^−1^	
	Cholesterol	Amperometry	0.36 μmol L^−1^	30–240 μmol L^−1^	[107]
	Choline		0.08 μmol L^−1^	0.5–4 μmol L^−1^	
	miR-1		0.31 pM		
	miR-208b	EIS	0.37 pM	0.1 pM–10 nM	[113]
	miR-499		0.77 pM		
Diabetes	Glucose	Amperometry	209 μmol		[114]
	Insulin		340 μmol		
	Glucose	Amperometry		0–200 μM	[111]
	Lactate			0–30 μM	
	Glucose	EIS	58 mg dL^−1^	50–800 mg dL^−1^	[115]
	L-tyrosine		0.3 μmol L^−1^	1–500 μmol L^−1^	
	I27L	ECL	8.1 × 10^−12^ M	1.0 × 10^−11^–1.0 × 10^−7^ M	[116]
	I27L	ECL	23 fM	0.0001–100 nM	[117]
	miRNA-124	DPV	0.65 fM	1 fM–100 nM	[118]
	miRNA-21	coulometry	17 fM	10^−8^–10^−14^ M	[119]
Cancer	miRNA-155	DPV	6.7 fM	0.01–1000 pM	[120]
	miRNA-122		1.5 fM	0.01–1000 pM	
Prostate Cancer	PSA	DPV		1–100,000 pg mL^−1^	[110]
	PSMA Interleukin-6 (IL-6)			1–10,000 pg mL^−1^	
				1–1000 pg mL^−1^	
	Platelet factor-4 (PF-4)			1–10,000 pg mL^−1^	
	miRNA-375	DPV		miRNA-375	[109]
	miRNA-141		0.01–10 μM	miRNA-141 PSA	
	PSA				
	Methotrexate (MTX)	DPV	35 nM	5–1000 μM	[121]
Leukemia	Lactate dehydrogenase		25 U L^−1^	60–700 U L^−1^	
	Uric acid (UA)		450 nM		
	Urea		20 μM		
Breast Cancer	miRNA-155	DPV	0.98 fM	1 fM–10 nM	[122]
	miRNA-21		3.58 fM		
	miRNA-16		0.25 fM		
	miRNA-155		0.33 fM		
	miRNA-21	SWV	0.04 fM	0.001–1000 pM	[123]
	miRNA-210		0.28 fM		
Rheumatoid Arthritis (RA)	Anti-CCP-ab	EIS	0.82 IU mL^−1^	1–800 IU mL^−1^	[112]
	CXCL7	Amperometry	0.8 ng mL^−1^	1–75 ng mL^−1^	[124]
	MMP3		1.2 pg mL^−1^	2–2000 pg mL^−1^	
SLE	BAFF	Amperometry	0.08 ng mL^−1^	0.24–120 ng mL^−1^	[125]
Colorectal Cancer	APRIL		0.06 ng mL^−1^	0.19–25 ng mL^−1^	

**Table 3 sensors-22-05200-t003:** Optical sensors for multiplexed detection of biomarkers for metabolic syndrome (MS).

Metabolic Syndrome	Biomarker	Optical Method	LOD	Linear Range	Refs.
Cardiovascular Diseases (CVDs)	Procalcitonin (PCT)	SPR	1.22 pg mL^−1^	10–10^5^ pg mL^−1^	[154]
	Myoglobin (MG)	SPR	<1 ng mL^−1^	1–25 ng mL^−1^	[155]
	Cardiac troponin I (cTnI)		<1 ng mL^−1^	1–25 ng mL^−1^	
	Interleukin-6 (IL-6)	Fiber-optic fluorescence	5 pM (0.12 ng mL^−1^)	5–500 pM	[156]
	B-type natriuretic peptide (BNP) Cardiac troponin I (cTnI) C-reactive protein (CRP) Myoglobin (MG)	Fiber-optic fluorescence	0.1 ng mL^−1^ 7 × 10^−3^ ng mL^−1^ 700 ng mL^−1^ 70 ng mL^−1^	0.1–1 ng/mL 0.7–7 ng/mL 700–7000 ng/mL 70–700 ng/ml	[157]
		Fiber-optic SPR	1.48 ng mL^−1^	1–1000 ng mL^−1^	[158]
	Interleukin-6 (IL-6)	Electrochemical	0.886 fg mL^−1^	0.1–1000 pg mL^−1^	
Prediabetes	Glucose	Fiber-optic SPR	Can be tuned by changing the microgel concentration	16 μM–16 mM	[159]
	Glucose	Microfluidics-enabled multi-scattering of light	110 nM	1–400 μM	[160]
	Lactate		240 nM	10–3000 μM	
	Glucose	Colorimetric	27.2 μM	0.0781–5 mM	[161]
	Lactate		29.6 μM	0.0391–2.5 mM	
Prostate Cancer		LSPR	100 fg mL^−1^		[162]
	PSA			50 fgmL^−1^–5 ngmL^−1^	
	PSA	SERS	0.46 fg mL^−1^	0.46 fg mL^−1^–478.93 ng mL^−1^	[164]
	PSMA		1.05 fg mL^−1^	1.05 fg mL^−1^–113.4 ng mL^−1^	
	hK2		0.67 fg mL^−1^	0.67 fg mL^−1^–466.23 ng mL^−1^	
	PSA	SERS	0.37 pg mL^−1^	1 pg mL−1–10 µg mL^−1^	[165]
	CEA		0.43 pg mL^−1^	10 pg mL−1–1 µg mL^−1^	
	AFP		0.26 pg mL^−1^	10 pg mL−1–1 µg mL^−1^	
	PSA	SERS	10 pg mL^−1^ for all proteins	-	[166]
	CEA				
	AFP				
Multiple Cancers	AFP	Silicon photonic sensor array	-		[168]
	ALCAM		-		
	CA15-3		-		
	CA19-9				
	CA-125				
	CEA				
	Osteopontin				
	PSA				
Rheumatoid Arthritis (RA)	miRNA-21	FRET	1 pM (both)	1 pM–1 nM (both)	[169]
	miRNA-155				
	miRNA-21	SPR	10 aM (both)	10 aM–10 pM (both)	[170]
	miRNA-155				
	miRNA-21	Silicon photonic Microring resonators	9 nM	20 nM–2 µM	[171]
	miRNA-26a		4 nM	20 nM–2 µM	
	miRNA-29a		<1 nM	2 nM–2 µM	
	miRNA-106a		2 nM	2 nM–2 µM	
	miRNA-222, miRNA-335		1 nM	2 nM–2 µM	
	let-7c-5p		4 nM	4–250 nM	
	miRNA-21	Silicon photonic Microring resonators	4 nM	4–250 nM	[172]
	miRNA-24		1.95 nM	1.95 nM–2 µM	
	miRNA-133b		62.5 nM	62.5 nM–1 µM	

Abbreviations: PSA, prostate-specific antigen; PSMA, prostate-specific membrane antigen; hK2, human kallikrein 2; CEA, carcinoembryonic antigen; AFP, alpha fetoprotein; ALCAM, activated leukocyte cell adhesion molecule; CA15-3, cancer antigen 15-3; CA19-9, cancer antigen 19-9; CA-125, cancer antigen-125; SPR, surface plasmon resonance; LSPR, localized surface plasmon resonance; SERS, surface-enhanced Raman Scattering; FRET, fluorescence resonance energy transfer.

## Data Availability

Not applicable.

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
