# Peer review of "Towards Multiplexed and Multimodal Biosensor Platforms in Real-Time Monitoring of Metabolic Disorders"

_sensors, 2022, doi:10.3390/s22145200_

Round 1
Reviewer 1 Report
This paper reviews the development of multiplexed and multimodal biosensor platforms in real-time monitoring of metabolic disorders (MS). In this review, various types of biomarkers related to MS and the non-invasively accessible biofluids available for sensing are summarized, mainly focusing on the two widely used sensing platforms, electrochemical and optical approach. This review needs to be revised carefully and hopefully the comments below will be able to help to further improve the review.
1. An abstract is often presented separately from the article, so it must be able to stand alone. Hence the problem statement, aim, and novelty and of the review have to be all included in. The author should also indicate the need for this review.
2. Graphical abstract: Not presented, it will be interesting for the author to present this review in illustration.
3. Introduction: The author should elaborate on the biosensors for MS detection as well. What is the general method used for the detection of MS and why is biosensor the focus of study in this review?
4. Section 2: The author could list the biomarkers for different types of MS into a table for easier referencing. Also, it would be beneficial if the author could also discuss the correlation of the biomarker concentration with the specific disease.
5. Section 2: Kindly refer to following paper that is relevant to the detection via biofluids:
Smartphone-based portable electrochemical biosensing system for detection of circulating microRNA-21 in saliva as a proof-of-concept
6. Table 1: I believe there are much more works done concerning the electrochemical detection of these diseases based on different types of biomarkers. The author should review more comprehensively to provide a clearer reference for the readers.
7. Section 3.2: The author should focus on reviewing the works done on optical biosensor for the detection of various biomarkers related to MS rather than a general introduction to the optical detection method.
8. The review is lacking in terms of illustration, which could make it easier for the readers to understand and comprehend the review.
9. The title suggests that this review discusses about the multiplexed and multimodal biosensing platform but throughout the review there was not discussion on the multiplex detection for the biomarker. As for multimodal platform, there also isn’t any work concerning the combined sensing methodologies of electrochemical and optical being discussed in this review.
Author Response
We have added a graphical abstract and a table of biomarkers for different types of MS to the manuscript. The suggested paper has been referenced and more works regarding electrochemical platform are listed. Also, additional information regarding optical biosensors for MS is described. Lastly, most of the platforms in this manuscript are multiplexed biosensors. However, there hasn't been that much report on multimodal (electrochemical + optical) biosensors for MS as to our knowledge. But we have included another multimodal work that we could find to the conclusion.
Reviewer 2 Report
The manuscript entitled Towards Multiplexed and Multimodal Biosensor Platforms in Real-time Monitoring of Metabolic Disorders describes various types of biomarkers related to MS, the non-invasively accessible biofluids for sensing, and electrochemical and optical MS sensing platforms. The overall presentation of the manuscript is fine. Conclusions are in line with the main text. However, revisions are needed before considering for publication.
1. Section 2 Biomarkers for Metabolic Syndrome: It is suggested that authors comment on the clinical level (stage) of the discussed biomarkers i.e., whether these biomarkers are clinically approved or not.
2. In section 2, genetic biomarkers are not discussed. There are many studies related to DNA or RNA (especially microRNA) biomarkers that should be considered and discussed in detail.
3. The subsection 2.5. Non-invasively Accessible Resources for Biomarkers seems not placed properly. It can be presented as a separate section (section 3).
4. “Here, the glucose level is higher than normal but is not considered to be 152 diabetes, classified as glucose level between 100 – 125 mg/dL (5.6 – 6.9 mmol/L)”. Statement needs citation.
5. Section 3.2. Optical Biosensors: This section needs to be revised. Authors should discuss multiplex optical detection systems in detail (principle, device setup, and data analysis).
6. Machine learning plays an important role in multimodal analyses. Authors may add a paragraph or a section to describe the role of machine learning in multiplexed biosensing.
7. Please add more insights in future prospects section. Comment on the application of CRISPR technology in MS biosensing.
8. Novelty of the work should be justified in light of published papers. Authors may consider some relevant literature: “The Need to Pair Molecular Monitoring Devices with Molecular Imaging to Personalize Health” doi: 10.1007/s11307-022-01714-4.
9. As the key word “Multiplexed and multimodal” used in the title, most of the contents are related to one modal technology like electrochemistry or optical biosensing probes, combination of electrochemistry and optical biosensing is rarely discussed, while it appears in the outlook parts. Please discuss it in the main text as well.
10. The text is hard to read in most of the figures.
Author Response
We have made a table of biomarkers for MS with its clinical stage in the manuscript and relevant genetic biomarkers are included throughout the review. The sections were rearranged with correct citations as well as the suggested literature. Optical biosensors have been revised to describe the multiplexed optical detection system in detail. CRISPR and ML has been mentioned in the future aspects. The figures have been modified for easier understanding. Lastly, there were not enough multimodal (electrochemical + optical) detection platforms for MS to our knowledge. However, we did mention additional multimodal platform in conclusion.
Reviewer 3 Report
This paper focuses on the relationship between metabolic syndrome (MS) and immune system dysregulation, in which the resultant biomarkers expressed have gained insight in the early detection of the associated metabolic syndrome (MS). However, sensing only a single analyte has its limitations, as one analyte may be involved in a variety of conditions. Therefore, a multi-analyte sensing platform is necessary for accurate diagnosis in MS, which usually results from the co-existence of multiple comorbidities. In this review, the authors summarized the various types of biomarkers associated with multiple sclerosis and the non-invasive biofluids that can be used for sensing. Two widely used sensing platforms, namely electrochemical and optical, are then discussed in terms of multimodal bio sensitivity, figure of merit (FOM), sensitivity and specificity for the early diagnosis of metabolic syndrome (MS). This provides a thorough understanding of the current status of existing platforms and how both electrochemical and optical modalities can complement each other to create a more reliable sensing platform for metabolic syndrome (MS). The article is adequate, well-structured and well thought out, but the following issues remain to be revised as suggested.
The following are the questions and some mistakes in this manuscript:
(1) The POC device is mentioned in line 86 and it is recommended to add a description of how the POC device works.
(2) The biomarker CRP for cardiovascular disease is mentioned in line 137, as well as the new biomarkers GA and OG mentioned later in line 169, which are terms that appear for the first time in the text suggesting that their full names be given.
(3) In line 141 it is mentioned that CRP, a biomarker of cardiovascular disease, is not an important factor in cardiovascular disease, but why is it highlighted in the text.
(4) In line 431 microdialysis is mentioned and it is suggested to add a schematic diagram of how it works.
(5) It is recommended that the table name of Table 1 be placed at the top of the table.
(6) Figures 1, 3 and 4 appear crowded and the text in the figures is too small.
(7) References 7, 12, 14, 21, 26, 33, 41, 47, 52, 58, 78, 79, 80, 95, 106, 117, 130 are missing doi numbers.
To conclude, this paper is well organized, but there are still many of the above problems that need to be revised.
Author Response
Abbreviations are now explained thoroughly throughout the manuscript. For " In line 141 it is mentioned that CRP, a biomarker of cardiovascular disease, is not an important factor in cardiovascular disease, but why is it highlighted in the text.", we wanted to indicate that although it is still relevant to CVD, CRP alone is not sufficient enough to be used for precise diagnosis of CVDs. The paragraph for microdialysis has been removed for better understanding. The names of the tables are moved to the top of the table and the figures have been modified for better readability. Lastly, the doi numbers of the references have been added.
Round 2
Reviewer 1 Report
It would be better if the author could response to the comment point-by-point and indicate the changes done.
Reviewer 2 Report
Ready to go.
Reviewer 3 Report
All are Revised, it can be published.